# The Sustainable Niche for Vegetable Production within the Contentious Sustainable Agriculture Discourse: Barriers, Opportunities and Future Approaches

Dickson Mgangathweni Mazibuko [1,2,*,†] , Hiroko Gono [3,†], Sarvesh Maskey [4,†], Hiromu Okazawa [4,*,†] , Lameck Fiwa [5], Hidehiko Kikuno [3] and Tetsu Sato [6]

1   School of Natural and Applied Sciences, University of Malawi, Zomba P.O. Box 280, Malawi
2   Graduate School of Agro-Environmental Science, Tokyo University of Agriculture, Tokyo 156-8502, Japan
3   Faculty of International Agriculture and Food Studies, Tokyo University of Agriculture, Tokyo 156-8502, Japan
4   Faculty of Regional Environment Science, Tokyo University of Agriculture, Tokyo 156-8502, Japan
5   Faculty of Agriculture, Lilongwe University of Agriculture and Natural Resources, Lilongwe P.O. Box 219, Malawi
6   SDGs Promotion Office, Ehime University, Matsuyama 790-8577, Japan
*   Correspondence: dmazibuko@unima.ac.mw (D.M.M.); h1okazaw@nodai.ac.jp (H.O.)
†   These authors contributed equally to this work.

**Abstract:** Agricultural productivity impacts the environment and natural resources in various ways. The severity of these impacts has triggered the emergence of natural resource management and the related, highly criticized science of agroecology. Vegetable production has known environmental impacts. However, the extent of its participation in sustainable production has not been adequately explored. This review sought to explore the spaciotemporal position of vegetables in a suite of existing sustainable agricultural practices, explore regional variations and discover lessons that can guide the future of vegetable production. There are regional differences regarding sustainable production practices and the associated barriers to their adoption. Generally, sustainable agricultural practices with a societal history in a region tend to be successful, unlike when they are "new" innovations. The major barriers to sustainable agricultural practices in vegetable production are economy-related (total investment cost) and crop-related and are also related to the technology transmission approaches. Unfulfilled expectations and a lack of community participation in technology development are noted challenges, which have led to dis-adoption. A farmer-centered approach to technology promotion could help. Comparatively, southern Africa has the most challenges in the adoption of sustainable agricultural practices. From the lessons learned from other regions, agroecology in vegetable cultivation is not unachievable in Africa. The projected challenges mean that sustainable vegetable production is inevitable.

**Keywords:** agroecology; agricultural technologies; community dialog; opportunities; sustainable natural resources; vegetable

## 1. Introduction

### 1.1. Agroecology and Natural Resource Management

Sustainable agricultural practices are currently being promoted as a method of adapting to climatic variability and as an alternative to feed the future human population. They are at the center of the promotion of modern-day agriculture. Today, agriculture is being hailed as a major force for poverty reduction [1,2], with vegetable productivity poised to play a key role in this endeavor [3,4]. However, McCullough et al. [5] argue that changes within the food systems necessitate new research into agriculture's role not only in improving people's socio-economic status but also in preventing its broader environmental impacts. Currently, most vegetable farmers are still trapped in the traditional food system with unorganized supply chains and limited (or non-existent) market infrastructure [5].

This notwithstanding, sustainable agricultural practices are still hailed as avenues for attaining food sovereignty [6] and improving the socio-economic wellbeing of farmers. Such agroecological appropriateness remains debatable.

### 1.2. Sustainable Agricultural Practices: A Brief History

Sustainable agricultural practices can be traced back to the origins of the conservation movement in the early 1900s, when the movement, launched in 1908 in the United States, was created as a response to the observed science- and technology-led "destruction of forests and wildlife, overgrazing and a too ambitious agriculture that produced deserts, as well as waterways that alternately flooded and ran dry" [7]. This movement led to the birth of agroecology and the associated "agroecological practices" [8]. In the 1960s and 1970s, agricultural impacts on the environment and natural resources became apparent, documenting early "ecotope" and "biocoenosis" [9]. Such impacts necessitated that the agricultural development trajectory be coupled with environment protection under the broader umbrella of natural resource management. Over time, the specific role of agricultural intensification on broader ecosystems has led to agroecological concepts. Agroecology as a term is shrouded in terminological confusion, as evidenced in reviews by Bell and Bellon [10] and Gomez et al. [11]. Agroecology has been defined in various ways, and its placement within the natural resource management literature is rather fluid. In this article, agroecology is broadly considered as a science dealing with practices under sustainable agriculture productivity, as shown in Figure 1. The various agroecological practices/technologies that interact with the agroecology philosophy are presented in Figure 1. These practices are widely referred to as sustainable agricultural practices. In this article, we use "sustainable agricultural practices" synonymously with "agroecological practices".

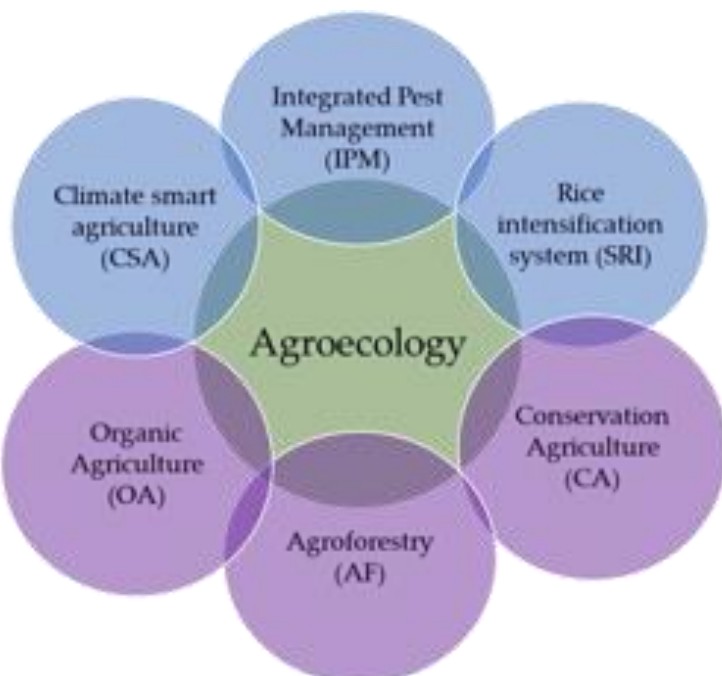

**Figure 1.** Conceptual overview of the various scientific approaches facilitating integrated resource management under the broader umbrella of agroecology and associated "agroecological practices". Modified from Ref. [12].

Agroecological practices have faced criticism from various sources. Critics have raised doubts regarding the high yield outputs [13], the ability to feed the growing human population [14], the impact of such practices on environmental conservation [15], their scalability capacity [16], the inadequate support for people's socio-economic improvements [17] and the questionable adoption trends of the practices themselves [18]. For Africa, the

criticisms are even more serious. Mugwanya [19] contends that most African agriculture is already agroecological in nature (thus, its promotion aims to continue with the existing system that has already failed Africa), that yields from agroecology practices do not surpass those from conventional agriculture, that agroecology is against agricultural modernization and that AE practices have been refined to perpetuate poverty among African farmers. Mugwanya [19] concludes by stating that "agroecology is a dead end for Africa". However, is this the case for vegetable production? Most of the challenges regarding the feasibility of sustainable agriculture are based on traditionally major agricultural crops. Here, it is suggested that sustainable vegetable production could equally be influenced by socio-ecological dynamics. Sustainable vegetable agricultural success could partly be dependent on the "crop–technology" mix and the relative importance attached to a crop (market and subsistence), among other factors.

Irrespective of the skepticism regarding the feasibility of sustainable agriculture both in general and in vegetable production, in this paper, it is accepted that barriers to adoption exist, yield outputs are practice-dependent, and therefore, not all sustainable agricultural practices are suitable for all agroecosystems [20] and crops. In this review, we argue for the feasibility of sustainable vegetable production and that, irrespective of the barriers and skepticism regarding sustainable agricultural practices, there exists a niche for vegetable production in a future facing climate change and the need to feed a growing human population. We first provide the reasons for sustainable vegetable productivity; secondly, an analysis of several sustainable vegetable production practices in China, Europe and southern Africa is presented; and thirdly, we provide a brief overview of barriers specific to sustainable vegetable production. This is followed by a documentation of working sustainable vegetable production practices and key lessons for emulation in southern Africa. Finally, this work offers a framework proposal on the key steps to encourage the adoption of sustainable agricultural practices.

## 2. Review—Methodological Overview

This review aimed to obtain an up-to-date, yet broad and generalized, scenario regarding the space vegetables occupy in sustainable agriculture. As such, the publications used were primarily based on two criteria. First, the articles had to have been published no earlier than 2015 (except where the article dealt with a narrative not recently published). Secondly, articles should deal with sustainable vegetable production practices in a broad sense (crop-specific articles were considered where broader publications on a particular issue were missing).

Review period: The target articles were those published between 2015 and 2022. This was to ensure that the latest knowledge, practices and debates were captured. The search results were customized to fit within the desired period in the search engines. Some key articles published earlier than 2015 were included (where no comparable research was available during the desired period). In total, two hundred and fourteen (214) articles were analyzed. Of these, 74% were from 2015, and 10% were published earlier than 2010.

Database search tools: The study used a combination of four search engines and databases. These comprised Google Scholar, a free web search engine, which indexes the full text or metadata of scholarly articles in diverse disciplines; AGRIS (FAO), a database indexing articles and other publications in food and agriculture; Scopus, Elsevier's abstract and citation database for life sciences and social sciences; and Mendeley, a reference manager with an article search option. The Boolean search approach was used for searching all the databases listed above.

Search terms: A variety of search terms were used to find relevant publications. First, a general search was conducted with terms such as "Agroecology, vegetables, China"; "crop rotation, vegetables, southern Africa". More refined searches were performed, identifying articles on barriers to adoption, e.g., "crop rotation, barriers, adoption". The words "vegetables" and "country name" were maintained while varying the word(s) for a given sustainable agricultural practice. This was repeated for all the ten practices chosen.

Choice of practices: This review focused on ten sustainable agricultural practices. These are: agroforestry, conservation agriculture, crop rotation, green manure, mulching, improved seed (and genetically modified seed), irrigation, intercropping, organic agriculture and precision agriculture. These practices were chosen to include those that are relatively widespread from the author's perspective. These practices also show a spread of input requirements (from relatively low to high) and those that have advanced in terms of technology.

## 3. Why Sustainability in Vegetable Production?

The urgent need for sustainable vegetable production stems from several issues. These include the fact that vegetable production uses some unsustainable practices; the need to satisfy the recommended daily vegetable intake; the dilemma of reducing agricultural landholding with the increasing human population; the maintenance of genetic diversity; and mitigating the vegetable production footprint.

*Unsustainability in vegetable production*: Vegetable cultivation has long been carried out along riverbanks/catchments. This practice is partly driven by the lack of access to productive land in proximity to a water source (rivers) and the need to supplement income for urban dwellers of a low economic status. Riverbank vegetable cultivation has consequently led to siltation and drying up of streams [21], thus negatively impacting other river ecosystem provisions [22]. Manure usage in vegetable cultivation has been found to lead to heavy metal contamination [23], eutrophication of river systems and the release of greenhouse gases. Other studies find that riverbank vegetable production aids climate change resilience and adaptation [24] in southern Africa. Due to soils being fertile, riparian vegetable cultivation will remain attractive, especially with the increasing human population. It is thus imperative to identify suitable technologies to make this agricultural enterprise sustainable. Unsustainability in vegetable production is not confined to southern Africa. In Europe, under protected agriculture, the pollution of natural water systems has been attributed to vegetable production [25,26].

*The need to bridge the malnutrition gap*: The production and consumption of vegetables are lower than the WHO recommended daily intake of 400 g/d. This has emanated from global underproduction and shifting dietary patterns of vegetable intake to some extent. The underproduction of vegetables is multifactorial, including limited access to land and input, the historical emphasis on staple crops, the lack of purchasing power and limited economic returns from their cultivation, among other factors. If the goal is to combat global malnutrition (especially micronutrient deficiency), efforts to boost non-market (consumption vegetable) cultivation need to be stepped up.

*Inadequate vegetable intake*: Currently, the vegetable intake requirement is 240 g/d, with a recommendation to increase it to 300 g/d. This increase leads to a need to sustainably increase vegetable supply by 75% globally to meet the associated demand [27]. In Africa, vegetable production and consumption are low, with only 13% of countries having a vegetable supply that meets the WHO recommendation of 240 g/d [28]. This means that vegetable and fruit availability and supply are too low to meet the demand based on WHO intake recommendations. For southern Africa, between 1960 and 2015, the vegetable supply has remained low, with only one country meeting the recommended intake. Future projections are dire for southern Africa even when vegetable and fruit waste is eliminated. The vegetable supply shortfall needs to be confronted in a multidisciplinary manner, where improved production is coupled with affordability. Changing the society's attitudes toward vegetable consumption globally, and particularly in southern Africa, is also worthy of effort.

*Land and population distribution in southern Africa*: Agricultural land area is reducing globally due to the increasing human population. Smallholder farm sizes are usually less than 2 ha in land area [29]. It is currently estimated that four to six people occupy a single hectare of farmland. This is expected to increase to 8–12 people per ha in 2050 [13]. Inevitably, this will lead to land conflicts and the expansion of land into reserved ecosys-

tems, among other things. Under land constraints, vegetable production needs to adopt sustainable approaches to ensure continued productivity. The distribution and possession seem to be evolving. For some time, most smallholder land sizes ranged between 0 and 5 ha. The current evidence indicates the emergence of medium-sized farms acquired by the middle class [30]. This scenario has several implications. First, it will further increase land scarcity for smallholders [31], leading to households with smaller landholdings as the disadvantaged population will sell land (to the emerging middle class) for immediate financial gain. Since farm size determines household food self-sufficiency [13], the latter scenario could spell an overall increase in food insecurity in rural parts of southern Africa. The emergence of medium-sized farms, however, entails an expansion of land fit for mechanization and other sustainable agricultural practices (which are otherwise unfit for small landholdings). Irrespective of future landholding dynamics, the practice of relevant agroecological practices remains an avenue, which guarantees sustained land productivity.

*Maintenance of genetic biodiversity*: The decline in biodiversity of plants that have supported humanity due to the onset of the green revolution [32] has particularly affected vegetable diversity. There is currently an urgent need to conserve local vegetable biodiversity [33]. To achieve this, the cultivation of local vegetables (sources of breeding diversity) needs to be promoted, albeit with less emphasis on economic returns emanating from their cultivation. The unpredictability, let alone availability, of markets for indigenous vegetables in rural communities risks the "for-profit" promotion campaign being a failure.

*Role in reducing agricultural footprint*: Real sustainable agriculture (especially where the intention is to mitigate climate change impacts) will require non-profit motivations. Over the years, the goal of community vegetable production has been subsistence. The promotion of "novel" sustainable agriculture practices implicitly aims to couple production with profit. While this is an ideal scenario, a meaningful impact on climate change will require practicing such technologies solely for the sake of participating in the "war" on climate change. Finally, the predicted dwindling farm sizes will simply require innovative (and at times costly) approaches to sustain vegetable production even for consumption alone.

## 4. Exploration of Diversity of Sustainable Vegetable Agriculture Practices

Early role of vegetables in agroecological practice: In agroecology development, vegetables have participated as enablers for achieving sustainable staple production. Further, vegetables have been used in intercropping systems to help in pest regulation of major staple or commercial crops. Altieri et al. [9] documented working intercropping systems of the 1960s and 1970s, where vegetables were not the focus of the sustainable practice but simply enablers. Today, the importance is given to vegetables, and their production is high. However, they are still not prioritized in land allocation, thus being produced in marginal lands of smaller sizes. This notwithstanding, sustainability in vegetable production stands to reduce the overall agricultural impact on the environment. With climate change impacting agriculture, vegetable production ought to be carried out in a climate-friendly manner. The discussion that follows pertains to a selection of agroecological practices in vegetable production in Europe, China and southern Africa. Vegetable production in developed Europe occurs across all farm sizes, from smallholder to large farms (2 ha to 200 ha). In China, 60% of vegetable cultivation occurs on land of less than 2 ha [29], and the country has severe water shortages per capita, placed 121 in the world [34]. Meanwhile, southern Africa is a representative region of the developing areas with agroeconomic challenges. Our aim is to extract lessons from documented barriers to sustainable agriculture practices in vegetable production in these regions. These shared lessons can provide a springboard for the enhanced sustainable vegetable production, which is required to meet the projected demand.

*4.1. Vegetable Status in Sustainable Agricultural Practices: Europe*

Agroforestry: In Europe, agroforestry has been a traditional aspect of landscape management [35] and is thus a mature and well-established practice. Vegetable production within these systems has been in practice across Europe [36]. In Greece, for example, agroforestry systems cover 23% of the country. Vegetables are classified as a major product, especially where *Prunus* sp., *Malus communis* and *Cydonia oblonga* are the dominant agroforestry trees [37]. Vegetable species grown in such agroforestry systems include chickpeas and common beans. The observation that 74% of agroforestry trees in Europe are broad leaved [38] and in Mediterranean environments is interesting. This observation provides a precursor for investigating candidate tree species, which can support productive agroforestry in other regions. Various aspects of European agroforestry and vegetables are dealt with in Refs. [39,40].

Conservation agriculture: As a practice, conservation agriculture has been comparatively less adopted [20], comprising only 1.2% of arable land [41]. Where it is practiced, conservation agriculture is farmer-driven, motivated by savings on machinery, fuel and labor, and not for soil and water conservation, as is the intention of conservation agriculture.

Crop rotation: Crop rotation in vegetable production maintains soil structure and organic matter, reducing resident soil pathogens while increasing nitrogen-fixing microbes. Crop rotation has also been shown to control soil erosion and increase biodiversity [42]. Due to these benefits, crop rotation is a common practice in organic agriculture, where artificial soil and disease control mechanisms are not allowed [43]. Crop rotation, however, is practicable where land size is not a constraint [44], and as such, it is usually practiced in commercial vegetable production systems.

Green manure: The use of green manure is prevalent in organic vegetable agriculture settings [45], especially in Mediterranean countries. Its usage in conventional (non-organic) production could not be verified.

Mulching: Mulching (green mulch) practice was first studied using vegetable production [46], and its use dates back to the 1970s. Because of its complexity, as described by Ref. [46], it is not popular as a sustainable agricultural practice among most farmers. In Europe, it is a widely adopted technology [47]. The current debate now surrounds the promotion of biodegradable mulch as opposed to traditional black polyethylene [48,49] and the promotion of their adoption due to cost limitations [50].

Improved seed: Improved vegetable seed/varieties is not a challenge in Europe compared with other regions, with the Netherlands representing a global leader in plant seed trade [51]. European farmers have access to quality seed from credible seed producers. These producers create hybrids, which cannot be multiplied by farmers [52], forcing vegetable farmers to rely on quality seed in every production cycle.

Irrigation: Drip and sprinkler irrigation technologies have been widely adopted as a standard vegetable production approach [26], especially in protected cultivation. The current challenge for Europe lies in the efforts to reduce extensive nitrate ($NO_3^-$) leaching and contamination of natural water resources [25,53] from vegetable production.

Intercropping: Mixed cropping with vegetables is an old practice, which is gaining prominence in Europe [54,55] and is associated with greater efficiency of land use and inputs [56]. This makes intercropping suitable for smallholder farmers. Further, intercropping aids in risk minimization, increased income and food security [57] and pest and disease control, improves soil fertility, improves product quality and generally enhances land use efficiency [58]. Such advantages position intercropping as a robust system in an era of dwindling landholding, population growth and climate change.

Organic vegetable: This kind of agriculture has been well established. Europe has the largest land (201,071 ha) area dedicated to organic vegetable cultivation [59] in the world. In Austria, for example, 35% of organic farms grow vegetables. Smallholder vegetable production in Europe is performed in greenhouses, which falls under "protected cultivation". Within protected cultivation, there exist sustainable agriculture practices, such as irrigation, mulching and precision agriculture. Zero tillage is a prevalent and working

technology and is a component of organic vegetable production both in Europe [60,61] and elsewhere [62].

*4.2. Vegetable Status in Sustainable Agriculture Practices: China*

According to Gliessman's study [63], agroecology and its principles/practices had not had a formal presence in China until 2016, when the "International Symposium on Agroecology for Sustainable Agriculture and Food Systems in China" was held. Before this, Chinese agriculture had been driven by the "rural-based" Chinese Ecological Agriculture (CEA) approach, launched in the 1980s [64]. Modern technologies, such as conservation agriculture, originated around the 1990s [65]. This notwithstanding, the literature on Chinese agriculture reveals that agroecological practices by farmers are in sync with those in areas where formal agroecological agriculture has long been practiced. China has a diverse genetic resource of vegetables, most of which have been conserved in GenBank across the country. There are 246 vegetable species from 50 families and 152 genera, which are grown as vegetables, in addition to 255 vegetable species (25 families and 44 genera) in the wild [66]. Many of these vegetables have been cultivated on smallholder farms.

Agroforestry: Hsiung and colleagues [67] traced agroforestry practice to as far back as 1600–800 B.C. Due to population pressure and industrialization, agroforestry has failed to maintain relevance [67], giving way to modern agroforestry. Vegetable gardens are at times located within the agroforestry system [68]. According to Ref. [69], agroforestry is not a key player in agricultural production in China, occupying only 1% of the studied area. Of this 1%, vegetables occupied even less land.

Conservation agriculture: China is ranked eighth globally in terms of conservation agriculture adoption [70], and it is a region where a policy-driven increase in agricultural productivity has been linked with increasing uptake of modern agricultural practices [71]. Research specific to vegetables is rather scarce.

Crop rotation: While traditional crop rotation can be traced back to 770–476 B.C. [72], modern crop rotation in China seems to be a new technology, with limited spread. Crop rotation and fallow system trials have only begun being developed between 2016 and 2021 [73], with national plans to expand the trial area and promote the diffusion and adoption of a crop rotation system [73]. The role of vegetables in crop rotation for China has yet to be documented.

Green manure: The use of green manure is not a common practice in vegetable cultivation in China, as evidenced from the literature deficiency. Green manure technology is, however, used extensively in rice [74], wheat [75] and maize [76].

Mulching: Mulching is a well-developed technology in China, but plastic mulching dates back to the 1980s [77]. It plays a critical role in vegetable cultivation [78,79]. The adoption of biodegradable mulching (as a sustainable agriculture practice) is, however, a challenge for smallholder farmers that rely on government subsidies [80], irrespective of them being the largest users of plastic mulching.

Improved seed: The use of F1 vegetable hybrids and the establishment of the vegetable seed industry have been well documented [81]. For genetically modified seed, Ref. [82] notes that China could be a potential market for genetically modified crops. The country has certified genetically modified crops for development and is receptive to GM-labeled imports. This could be the case due to the level of debate (and associated awareness) concerning GM products not having reached the level that it has in Europe.

Intercropping: Intercropping is a very old practice in China, which has been in practice for thousands of years [83] and is comparatively widespread while being in decline in other areas [84]. Intercropping continues to provide the pathways for ecological intensification of agricultural food production.

Irrigation: Irrigation agriculture in China dates back to 598 B.C. [85] as a practice embraced due to persistent droughts, especially in the northern part of the country [86]. The irrigation technology uses ground water, a source that is being depleted [85,86]. Vegetable production has always been a part of this bigger picture. Recently, however, modern

irrigation technologies have been introduced in agriculture [34,87], both generally and in vegetable production.

Organic vegetable: Organic vegetable production in China is a well-developed technology. China is ranked third in terms of land dedicated to organic agriculture, 0.2% of which is allocated to vegetables [59]. Like most sustainable agriculture practices, organic agriculture is claimed to date back 4000 years [88]. Smallholder farmers, grouped into "farmer groups" or "farmer organizations", are behind most of the organic agriculture productivity in China [88].

Precision agriculture: Precision agriculture as a practice in agriculture is relatively new. The current research is focusing on adoption dynamics and methods of enhancing it [89,90]. Its application in vegetable production has not yet been documented.

*4.3. Vegetable Status in Sustainable Agriculture Practices: Southern Africa*

In southern Africa, some sustainable agriculture practices have a historical background. Generally, sustainable agriculture practices are still faced with adoption issues, for which various solutions have been suggested [91,92]. Here, we highlight sustainable agriculture practices specifically in terms of vegetable production in southern Africa.

Agroforestry: Research into vegetable agroforestry is scarce. This could be indicative of limited research in the area emanating from a lack of practice in southern Africa. This notwithstanding, the roles and feasibility of agroforestry in promoting vegetables have been recognized [93] along with agroforestry's potential contribution to human health in southern Africa. However, home gardens, a form of agroforestry, have been proven to increase commercial vegetable cultivation [94].

Conservation agriculture: This is the premise of three soil improvement practices: minimum soil disturbance, residue retention and crop rotation. These practices are rarely implemented as a package, with various combinations of the three being adopted in different countries (see Ref. [95]). Conservation agriculture has been adopted the most in Uganda and Ethiopia and the least in Mozambique. The reasons for non-adoption vary between households and with land size, albeit with conservation agriculture practices being adopted. Generally, small land size promotes non-adoption of crop rotation practices [95]. The other factors include a lack of land security and unavailability of extension.

Crop rotation: Research and publications on vegetable crop rotation in the region are rare. In the past, smallholding farmers used crop rotation and fallow systems to restore soil quality due to the lack of capital to acquire agricultural inputs. However, such a strategy is no longer feasible due to an increase in population and subsequent growing demand for food [96]. The land issue is worrying in the bigger picture, considering that small land sizes are usually allocated for vegetable production. Further, a lack of adoption of the practice could be exacerbated by the lack of exposure to the benefits of crop rotation practices.

Green manure/cover crops: The use of green manure and cover crops is common in organic vegetable production systems in developed countries [97]. However, this technology for vegetable production in southern Africa has not yet been widely documented. In Nigeria, green manure was proven to improve the yields and nutrient composition of tomatoes [98]. The use of green manure/cover crops has, however, been suggested for the African region due to the benefits for soil properties and weed suppression [99].

Mulching: Mulching in vegetables is a common practice in most African countries. It is commonly used to minimize soil evaporative water loss on seedbeds and protect seedlings from direct light following transplanting. Like other sustainable agriculture practices, the literature on this practice being used with vegetables is limited.

Improved vegetable varieties: Vegetable production in southern Africa refers to both introduced and indigenous vegetables. Most introduced vegetable seeds make use of improved seeds (via hybridization). For indigenous vegetables, the availability of quality seeds is a challenge, except for *Solanum aethiopicum* [100] and *Amaranthus* sp. [101]. For African indigenous vegetables, the challenge of using quality seeds occurs as farmers still rely on retained seeds of landraces [102]. Currently, community-led efforts to improve

indigenous vegetable seed quality and accessibility have been tested [103]. For African indigenous vegetables, while the supply of improved seeds is steady, the cost does limit some smallholder farmers [104]. The use of improved seeds as a sustainable agriculture practice is thus still limited.

Intercropping: Though a historical practice in Africa [105], intercropping has become more relevant due to a decline in landholding sizes resulting from population growth [96]. Traditionally, intercropping involved staple foods rather than vegetables. The past involvement of vegetables in intercropping was as a "protection crop" against pests of the majority crop (corn, wheat, cotton), as documented in Ref. [9]. Today, however, the role of vegetables in intercropping needs to evolve in order to support the growing demand for nutritious foods. Cowpeas and pigeon peas are two vegetables involved in intercropping [106], albeit this being due to their nitrogen-fixing abilities.

Irrigation: Traditional surface irrigation using buckets [107] and later watering cans in southern Africa in dry-season vegetable production has been a practice for a long time. Lately, drip irrigation of vegetables has been tested and found to positively correlate with yield. Yet, its adoption rates remain low [108] due to the high initial capital costs and its infeasibility on the smaller pieces of land (less than 0.4 ha) [109], which characterize vegetable cultivation in southern Africa.

Organic Agriculture: Vegetables are significant participants in organic agriculture. While all countries in south-eastern Africa (SEA) have some land under organic production, organic vegetable production has only been developed in South Africa, Kenya and Uganda, where 0.5%, 2.6% and 2.4% of their organic land is used for vegetable production, respectively [59]. This is reflective of the global situation, where only 0.7% of all organic land is used for vegetables. Due to its labor-intensive nature, organic agriculture is more likely to be adopted in small farms (with reduced labor needs), manageable using the joint labor of a family [110].

## 5. Barriers and Opportunities for Sustainable Agriculture Practices

### 5.1. A Synoptic Representation of the Barriers to Adoption Surveyed by Area

Critics of sustainable agriculture state that the adoption rates for such practices are lower than reported, that there are no short-term benefits and that some practices are not viable on a large scale [13]. Others have pointed to a lack of universal feasibility of some practices [111], as success can also be determined by socio-political considerations [112] for a given farmer or community, and unrealized expectations can lead to unsustainability [113], considering that vegetables have unique requirements regarding the production practices, land and growth. The possibility of variability in the barriers to adoption can thus be anticipated. Tables 1–3 below provide the documented barriers to the adoption of various sustainable agriculture practices in the three chosen regions.

**Table 1.** Barriers and challenges in Europe's sustainable vegetable production.

| Sustainable Agricultural Technology (Practice) | Barriers/Challenges to Adoption | References |
|---|---|---|
| Agroforestry | Increased labor, complexity of work, management costs and administrative burden. | [36] |
| Conservation agriculture | Lack of knowledge, information and communication about the practice; lack of enabling policies; lack of subsidies and credit. Crop-related factors include increases in weeds, pests, disease and pressure; crop failure; lack of skills; and low nutrient availability during key crop growth stages and management of weed pressure. | [20,114,115] |
| Crop rotation | Climate and soil limitations; low growth; lack of adapted crop varieties; and general market conditions. | [20] |
| Green manure | Cost of seeds; increased labor needs; competition with other crops; weed infestation. | [114] |
| Mulching | Cost of purchase and installation of equipment; difficulty in harvesting; labor constraints; rapid degradation of mulching and doubts about agronomic performance. | [47] |

**Table 1.** *Cont.*

| Sustainable Agricultural Technology (Practice) | Barriers/Challenges to Adoption | References |
|---|---|---|
| Improved seed (and GM seed) | EU policy and online articles questioning the safety of the method (among others). Genetically modified vegetables are thus not commonplace. | [116,117] |
| Irrigation | Income-related barriers include: lack of subsidies and access to credits; initial costs; output price; and water challenges, i.e., source of water, water price and its allocation. Other barriers include: farm size; land ownership; type of crops grown; technology complexity; and lack of communication of quality information. | [118,119] |
| Intercropping | Hinderance to mechanization and non-applicability to market demands. | [54] |
| Organic agriculture | Technical challenges; labor requirements; fear of decreased income and marketing problems; small farm size. | [120] |
| Precision agriculture | Income-related barriers include: high initial investment costs; unclear added value; too expensive and complex to use; and small farm size. Technology-related factors include: devices that are not interoperable and not precise enough and are unsuitable and unnecessary for smaller farms; lack of skills/capability required to adopt precision agriculture; reliability issues; knowledge intensity; and lack of perceived benefits. Other factors include: lack of neutral advice; lack of farm demonstrations regarding farmers' protection from risk and limited returns on investment. The practice is common among vegetable growers, but farm size limits broader implementation. | [121,122] |

**Table 2.** Barriers and challenges in sustainable vegetable production in China.

| Sustainable Agricultural Technology (Practice) | Barriers/Challenges to Adoption | References |
|---|---|---|
| Agroforestry | Lack of farmer interest; lack of sufficient knowledge; lack of capital; and lack of technical advice. | [123] |
| Conservation agriculture | Traditional attitudes; insufficient research and extension; lack of machinery tailored to conditions in China; competing usage of straw/residue; and site specificity. | [65,124] |
| Crop rotation | Not much documentation of barriers, since crop rotation is already extensively adopted as a fertility maintenance practice. The incentives provided may also have enhanced wider adoption. | [125,126] |
| Green manure | Key barriers include farmer's income, area of farmland and labor intensity. | [127] |
| Mulching | Biodegradability of plastic mulch. | [128] |
| Improved seed (and GM seed) | Active breeding programs are underway; the adoption (and barriers) of improved varieties is barely documented. Traditional breeding is prevalent, as is preserving traditional vegetable landraces. | [129] |
| | China has two genetically modified vegetables (tomato and sweet pepper). There is a lack of reliable information on genetically modified crop technology. | [130] |
| Irrigation | There is a lack of extension services; farm size may be wrong; there is water scarcity; there is a high investment cost; and there are high labor demands. | [44] |
| Intercropping | Limitations in mechanization with intercrops and lack of labor (since the practice is labor intensive). | [131] |
| Organic agriculture | Fear of risks from reduced yields; extra costs of certification of produce; intensive labor and unavailability of natural inputs in some places. Where adoption occurred, it involved "arm twisting" by local officials. | [132] |
| Precision agriculture | High investment cost, which favors large farms; incompatibility of software and hardware from different PA manufacturers; and knowledge intensity and need for quality technical support. Kendall and colleagues [133] comprehensively reviewed general barriers, which are equally applicable to vegetable production. | [133] |

**Table 3.** Barriers and challenges in sustainable vegetable production in southern Africa.

| Sustainable Agricultural Technology (Practice) | Barriers/Challenges to Adoption | References |
|---|---|---|
| Agroforestry | Barriers include: status of land tenure; small land size; limited access to credit; high investment costs; lack of knowledge and extension services; shortage of land; delayed returns on investment; and lack of seeds. | [134,135] |
| Conservation agriculture | Barriers include: small farm size; risks and uncertainties; high labor requirements; high initial costs; lack of local relevance; lack of skills; cash constraints; lack of equipment; limited availability and competition for crop residues; relative underperformance of conservation agriculture; low returns on investment and maize subsidies. | [136,137] |
| Crop rotation | Barriers include: farmer preference for food (cereal crops) over rotational cash crops; the unavailability of seed; dysfunctional markets for rotational crops; differences in planting techniques; plot size and land limitations. | [92] |
| Green manure | Barriers include: limited access to certified seeds; reduced diversity and lack of knowledge on productivity across agro-ecological zones; some inhibitive land tenure systems for long-term crops; high labor demand; lack of access to credit for inputs; lack of other uses for cover crop; cover crops hosting pests; and lack of specialized seed systems. | [138] |
| Mulching | Barriers include: lack of contact with extension workers; land tenure and ownership constraints; and labor-intensive practice. | [139,140] |
| Improved seed (and GM seed) | Barriers include: lack of awareness; lack of access to affordable seed; legal and political barriers; limited access to extension services; small farm size; and low farmer education. | [141,142] |
| Irrigation | Barriers include: a high price of irrigation kits; lack of access to credit; marketing challenges; lack of knowledge about drip irrigation technology; lack of adequate land; increased labor demands; small farm size; and seasonal scarcity of manure. | [143,144] |
| Intercropping | There is hinderance to mechanization, which is therefore less frequent on large commercial farms. On small farms, most intercropping involves a maize–bean mix. | [145] |
| Organic agriculture | Barriers include: comparatively lower yields; difficulties with produce certification; market barriers; and high farmer educational and research needs. | [146] |
| Precision agriculture | Technology is at the experimental stage in most countries. Where it is being tested, farmers decry: the lack of information; high cost of technology; small farm sizes; and low return on investment. | [147,148] |

### 5.2. Comparative Barrier Analysis

The barriers or constraints related to the adoption decision have been widely summarized, including by Adnan and colleagues [149]. These broader drivers of adoption manifest as various reasons for the adoption or non-adoption of various technologies. The barriers to sustainable vegetable production were categorized into: lack of knowledge, labor demand, income-related factors (equipment maintenance cost and lack of access to credit), equipment, farm status (size and tenure), crop-based factors (yield and seed access), lack of extension support and enabling policy. Figure 2 shows the frequency of references to these barriers from the reviewed articles. Labor, income needs and crop-related factors are common across the three regions. Overall, studies about China report the fewest barriers, whereas southern Africa reports the most. Farm size is the most often mentioned barrier to the adoption of sustainable vegetable production in Africa. Of interest is the lack of policy challenges in sustainable vegetable agriculture in China. This could mean it is not an issue in vegetable production, but it could be an issue when it comes to the production of other crops.

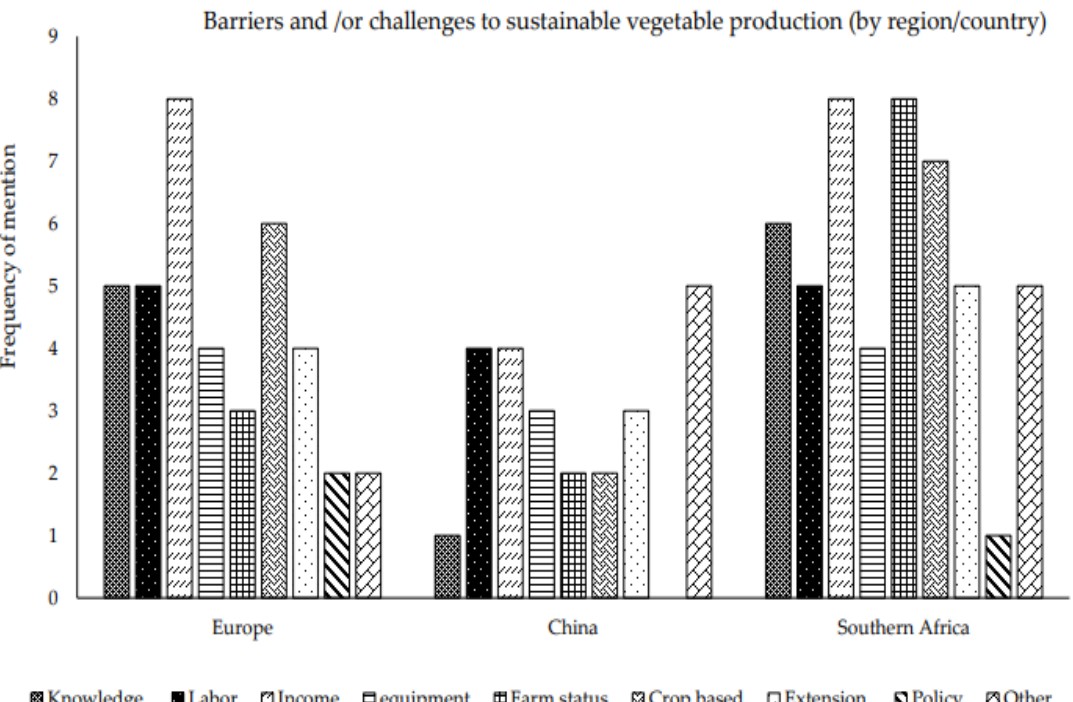

**Figure 2.** Categorization of various barriers to sustainable agricultural practices in vegetable production for Europe, China and southern Africa.

The most documented challenges are income- and crop-related factors. Income-related factors can be dealt with in various ways, including via subsidies and direct financial injections. Crop-related constraints can be managed via proper ecological and scientific screening depending on the socio-ecological status of the target area. The use of indigenous vegetables can, however, provide much support in addressing most of the crop-related concerns.

## 6. Lessons for Southern Africa and Developing Societies

Generally, there is a documented lack of adoption of sustainable agricultural practices in southern Africa [150]. The driving challenges highlighted above could mean a dark future for agroecology in this region. These documented lessons, while not prescriptive, could be valuable for some farmer trials. One major barrier of land size is bound to exist. Currently, 72% of the global food supply is being harvested on small farms [151]. Innovation aimed at sustainably maximizing outputs from the current land holdings is key for future agricultural productivity and particularly vegetable production.

### 6.1. Lessons from Europe

Conflict of interest impacts and agroecology technology promotion: In a research assessment of challenges in the adoption of smart farming technologies, two key responses were noted: "lack of neutral advice" and "lack of added value" [152]. These responses implicitly suggest that the technology producers were involved in its promotion as well. This conflict of interest stems from technology being developed without farmer consultation and without a clear problem to solve. In southern Africa, this mirrors the technologies promoted by funding agencies who have pre-packaged technologies to be adopted by farmers. Such technologies are bound to have low adoption levels; they may not respond to the challenges that farmers have identified.

Technological gaps and the need to be realistic: Regarding some technologies that are offered to farmers, while they may have had a long history in other societies, they could be novel to others. Some technologies, such as precision agriculture, while relevant [153], may

not be a priority to most farmers, especially in southern Africa (see Refs. [154,155]), where the adoption of basic sustainable agriculture practices is already a challenge.

Promotion of smallholder participation in markets' supply chains: One way to enable agriculture to participate in the drive for poverty reduction is to facilitate access to stable and reliable vegetable markets. In Europe, retail chains are widely linked to small-scale vegetable farmers [156], so this was initially expected. This worked for several reasons; small-scale farming provides the majority of the supply base, and it has a lower cost of labor, offers less complicated contracts and has reduced dependence on a few specific suppliers [156]. A surprising observation is the increase in productivity with a shift from labor-intensive large-scale farming to smallholder farms (0.2) people per hectare [157] in eastern Europe [156]. This is a positive lesson for southern Africa's smallholder vegetable farmers.

Incentives and subsidies within sustainable agriculture policy: The EU has the Common Agricultural Policy (CAP), which offers direct payments and subsidies to large farm food producers, and the Small Farmers Scheme (SFS) [158]. Eligibility for CAP payments is in part linked to having production that protects the environment through maintenance of farmland biodiversity and reducing greenhouse gas emissions, under CAP's greening Pillar I [159]. Such incentives are key, especially where agricultural practices are linked to climate change mitigation within a nation's policy framework.

### 6.2. Lessons from China

Farmer-led and problem-based technologies: In the case of intercropping, in Hebei province in China, the intercropping systems are created by farmers [131] in one area, which are then promoted in other areas. While this may not be appropriate for some technologies, it is a key approach for promoting adoption. Until late last century, China's agricultural production was not under formal AE influence. This notwithstanding, agroecological innovations are evident. Most of these innovations were problem-/challenge-driven [160–162], where societies noticed a crisis and innovated a solution. One key observation in China's approach to sustainable agriculture practice implementation is the presence of a "clear rationale" of farmers for each intervention. Such rationales are documented for irrigation [163], crop rotation [160] and green manure [161,162], where China ranks third globally in organic vegetable production [164].

Industry involvement promoting vegetable cultivation: Vegetables from organic agriculture have a comparatively higher price due to associated premiums. In China, some companies are involved in the creation of "organic villages" [165] where smallholder farmers grow vegetables organically that are then bought by a company. The same concept was replicated in India in 2006 [166]. With inherent logistical issues and certification of organic agriculture produce dealt with, this concept could be key in advancing rural income from vegetable production.

Utilization of alternative water sources (groundwater and wastewater): Water for vegetable cultivation can be a limiting factor for rural smallholder farmers. Such farmers depend on rivers or streams (most of which are becoming seasonal) during the dry season. To ensure year-round vegetable supply, in China, groundwater extraction for vegetable irrigation [167] is a common practice for smallholder farmers, as is vegetable irrigation using wastewater. Currently, however, vegetable contamination with heavy metals has raised public health concerns [168]. If cautiously promoted in SAA, such interventions could help with vegetable production.

Maintenance of feasible/working cultural practices: Each culture has certain unique food production practices. While some are being newly promoted today as part of AE technologies, they have long cultural histories in some cultures. For example, the use of green manure has a 300-year history in China [162], and organic agriculture dates back to the 1930s [165]. Today, such a culture fits as a sustainable AE technology. In Malawi, for example, 90% of cowpeas were produced from intercropping [54]. The lesson here is to

identify and promote the existing feasible practices in vegetable production and harness their linkages to "modern" scientific technologies. Such practices can easily be adopted.

Enabling national policy: China has enacted some policies, which are enabling the promotion of vegetable production. One example is the 2015 National Planting Green Manure Policy, aimed at promoting the use of green manure [127], and the National Land Consolidation and Rehabilitation Plan, which aims to increase cultivated land to 16,000 km$^2$ [169], among others. Such policies ensure sustainability in production. In Europe, there are regulations guiding the adoption of sustainable agricultural practices, which are mandatory [170] for most regions. This may thus influence the adoption determination. The introduction of such policies could be very useful in the promotion of vegetable cultivation.

### 6.3. Documented Sustainable Vegetable Production Practices

The adoption and spread of agroecological practices have not been consistent. The reasons behind this uneven spread are diverse and complex. Other scholars have blamed the translation of agroecological principles into practice [171]. In countries where agroecology was adopted early, such as Brazil, Mexico, India and countries in central America, agroecology is in advanced stages [172], while it is still lagging elsewhere. In part, these discrepancies could be explained by differences in the time of introduction, the non-linear distribution of agroecological crises that necessitate agroecological intervention and unequal familiarity of farmers with the practices, and agroeconomic dispositions also likely affected even adoption and spread. The eco-specificity of some technologies means that while a technology thrives in one region, it struggles in another. Above all, however, is the possibility that some technologies have not yet been trialed in ecosystems where they would thrive. The declaration of agroecological death in Africa [19] and general skepticism [13] need to be re-examined in the context of success stories in vegetable production. The success and "failure" of agroecological practices thus ought to be contextualized. There are sustainable agricultural technologies that work, and they work to varying degrees (Table 4).

**Table 4.** A collection of some working sustainable agricultural technologies in vegetable production.

| Sustainable Agricultural Technology (Practice) | Vegetable Taxa | Key Findings | Country | Reference |
|---|---|---|---|---|
| Agroforestry | Water spinach (*Ipomoea aquatica*), Malabar spinach (*Basella alba*) Amaranthus spp., Okra (*Abelmoschus esculenta*). | Reduction in vegetable yield under tree conditions. However, the yield indicates that the vegetable is still profitable. | Bangladesh | [173] |
| | Eggplant (*Solanum melongena*), Tomato (*Solanum lycopersicum*) and Chinese parsley (*Coriandrum sativum*). | Variable results. Generally better growth/productivity with increasing distance from tree base. | Bangladesh | [174] |
| | Chili (*Capsicum annuum*), eggplant (*Solanum melongena*) and Okra (*A. esculenta*). | Okra gave the highest yield under shade treatment. This was recommended for agroforestry systems. | Bangladesh | [175] |
| | Tomato (*S. lycopersicum*), brinjal (*S. melongena*), bhendi (*A. esculentus*), cluster beans (*Cyamopsis tetragonoloba*) and vegetable cowpeas (*Vigna unguiculata*). | *Solanum melongena* (brinjal) performed better under agroforestry with *Ailanthus*. Overall results for vegetable performance are variable. | India | [176] |
| | Irish potato (*Solanum tuberosum*), cabbage (*Brassica oleracea* var. capitatata), beans (*P. vugaris*), peas (*Pisum sativum*), wild strawberry (*Fragaria vesca*) and red raspberry (*Rubus idaeus*). | Import substitution by agroforestry community gardens (AFCGs) as socio-ecologically and culturally sustainable means of enhancing food security is feasible. | Canada | [177] |
| Conservation agriculture (zero tillage) | Tomato (*S. lycopersicum*) and lettuce (*Lactuca sativa*). | No difference between tillage and zero tillage in terms of yield (under optimal irrigation and fertilizer). | Australia | [178] |
| | Mustard (*Brassica* sp.). | Working technology in reduced-moisture environments. | India | [179] |
| | Lentil (*Lens culinaris*) and garlic (*Allium sativum*). | Improved energy efficiency in production of both crops. | Nepal | [180] |
| | Cabbage (*B. oleracea*) and brinjal (*S. melongena*). | This system was implemented. It showed that the system improved soil properties. | Brazil | [181] |

**Table 4.** *Cont.*

| Sustainable Agricultural Technology (Practice) | Vegetable Taxa | Key Findings | Country | Reference |
|---|---|---|---|---|
| **Crop rotation** | Kidney beans (*P. vulgaris*), mustard (*Brassica* sp.) and cowpeas (*V. unguiculata*). | Vegetable productivity was attained in some rotation set-ups (not all). | China | [160] |
| | Onion (*Allium cepa*) and sweet potatoes (*Ipomoea batatas*). | Demonstrated benefits of a "sustainable" rotation where potatoes or onions were included. | New Zealand | [42] |
| | Onion (*A. cepa*), lettuce (*L. sativa*), peas (*Pisum sativum*) and beans (*P. vulgaris*). | Onion, lettuce and strawberry were profitable under the cropping system. | USA | [182] |
| | Broccoli (*B. oleracea* var. italica) and cowpeas (*V. unguiculata*). | Cowpeas in rotation are good for crop diversification, reducing dependency on mineral fertilizers when growing broccoli. | Spain | [183] |
| **Green manure and cover crops** | Green beans (*P. vulgaris*), squash (*Cucurbita pepo*) and peppers (*Caspicum annuum*). | Cover crops improved soil biological properties and yields. The practice was found to be better for vegetable production, especially for organic farmers. | USA | [184] |
| | General vegetable assessment. | Cover crops uncommon in vegetable production. | USA | [185] |
| **Mulching (as part of conservation agriculture)** | Broccoli (*B. oleracea* var. italica), chili (*Capsicum annuum*) and garlic (*Allium sativum*). | Treatments of biodegradable mulch films (BDMs) and polyethylene mulch films (PEMs) effectively increased broccoli, chili pepper and garlic yields. | China | [186] |
| | Water spinach (*I. aquatica*). | Production of water spinach was significantly improved with rice straw mulching. | China | [187] |
| | Peppers *Capsicum chinense* and *Capsicum frutescens*. | Mulching plus reduced irrigation worked in improving yields. Ideal as a water conservation strategy. | Ghana | [188] |
| | Tomato (*S. lycopersicum*). | Mulching improved tomato yield (comparable to when herbicides were used). | USA | [189] |
| **Improved seed (and GM seed)** | GM tomatoes. | GMO safety certificates. | China | [190] |
| | *Amaranthus* sp. | Very high adoption, and the vegetable performance is good. | East Africa | [101] |
| | General vegetable assessment. | Genetically modified seed approvals in Europe are mostly pending. Research into most vegetables and fruits has already been conducted. | Sweden | [191] |
| **Irrigation** | Garlic (*A. sativum*), onion (*A. cepa*), tomato (*S. lycopersicum*), cabbage (*B. oleracea*) and sweet potato (*I. batatas*). | Drip irrigation for vegetables in home gardens was found to be a feasible strategy to improve water use efficiency and to intensify crop yield. | Sub-Saharan Africa | [192] |
| | Chinese cabbage (*Brassica rapa*), *Amaranthus* sp., tomato (*S. lycopersicum*), spinach (*Spinacia oleracea*), peas (*P. sativum*) and beans (*P. vulgaris*). | The strategy has the potential to improve farmers' resilience to climate change. The study found no evidence of poverty reduction. | Tanzania | [193] |
| **Intercropping** | Chili (*C. annuum*), garlic (*A. sativum*), onion (*A. cepa*), spinach and other vegetables. | Intercropping systems were developed by farmers and only promoted and spread by government workers. | China | [131] |
| | General vegetable assessment. | A comprehensive review of some working vegetable intercropping systems. | India | [194] |
| | Onion (*A. cepa*), cabbage (*B. oleracea*) and carrot (*Daucus carota*). | Carrot and cabbage can be sustainably grown with faba beans in an intercropping system. Faba beans have a positive influence on soil biological properties. | Latvia | [195] |
| **Organic agriculture (organic manure)** | Peas (*P. sativa*), faba beans (*Vicia faba*), cabbage (*Brassica* sp.) and radish (*Raphanus sativus*). | Harvested vegetables and plant remains are part of the green manure. | China | [162] |

## 7. The Future of Sustainable Vegetable Production: A Proposal

*Farmer-centered framework for sustainable agriculture practice dissemination*: One of the reasons for challenges to adoption is inherent in the way technologies are introduced to farming communities. A technology that clearly solves a farmer's "accepted" problem should be easily adopted. We argue that the discussed challenges to adoption (and consequent dis-adoption) are indicative of gray areas in the approach to technology dissemination. One possible scenario is where farming communities are "sold" a technology, usually with glorified outcomes, without farmer input of what challenge the technology intends to address. It is thus key that communities play a key role in deciding which technology could suit their agroecological circumstances. Emphasizing the lack of community dialog regarding drip irrigation technology in Burkina Faso, Gross and Jaubert [144] state: "Information is generally lacking as to who are smallholders, and what are their needs and constraints, yet development organizations devote little attention to answering these questions in their areas of intervention". Sulifoa et al. [196] report on community rejection of the imposed Mucuna pruriens as a cover crop (in conservation agriculture) by locals who instead preferred the local crop, Erythrina variegata. These examples highlight the importance of collective problem identification and solution co-creation. Recognition by

technology promoters of existing local technologies, an understanding of the key and "accepted" silvicultural problems and the selection of suitable crops for specific interventions should ensure active farmer involvement. In Figure 3, a suggested eight-step framework for sustainable agriculture practice dissemination is shown, where the extension–farmer dialog takes a central role.

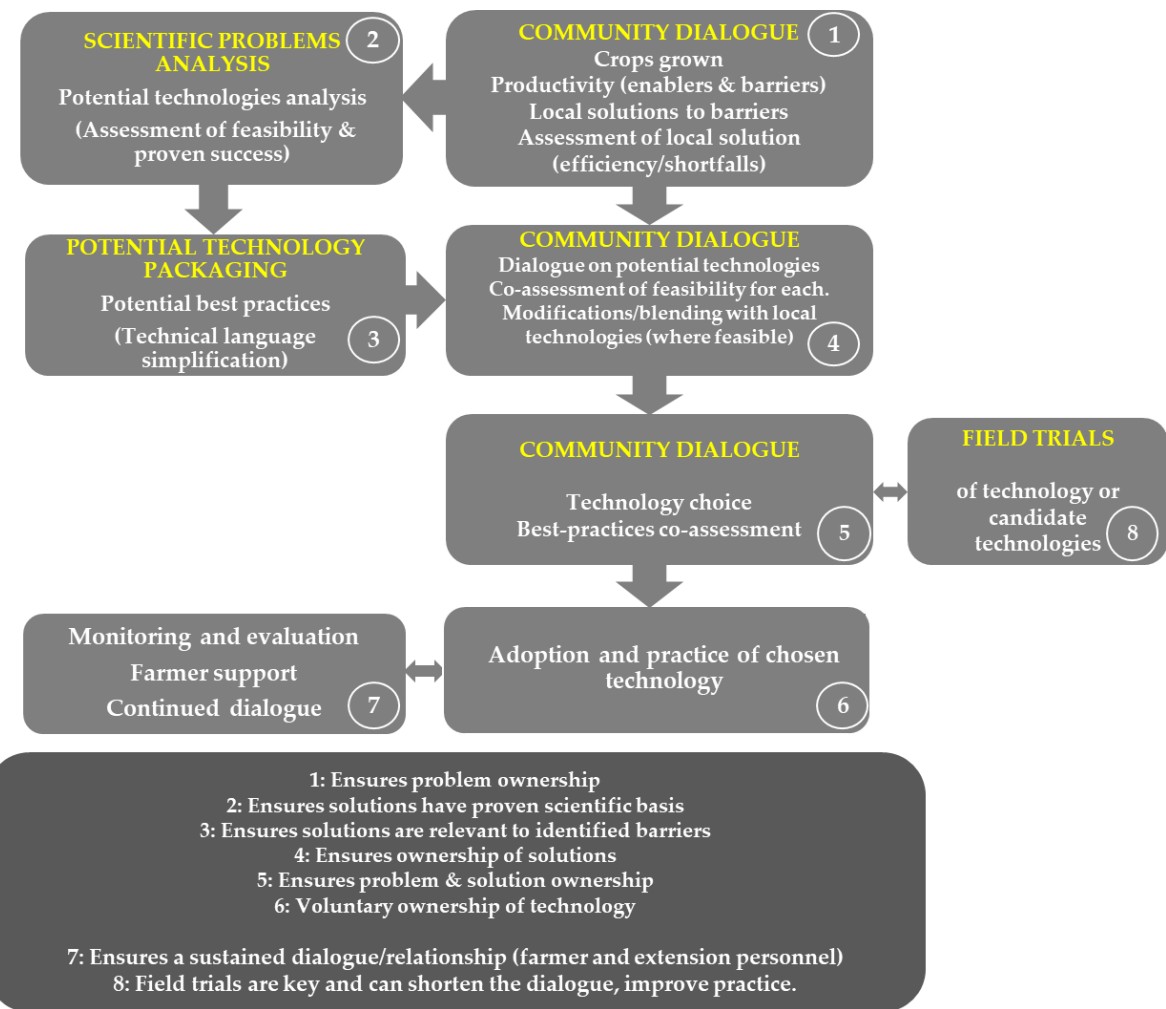

**Figure 3.** A suggested conceptual framework for sustainable agricultural practice dissemination and promotion in communities.

Key to the proposed framework is the emphasis on field trials and community participation. Local communities interact with natural resources directly in their everyday life. It is thus reasoned that the success of any integrated natural resource management strategy (be it sociological or scientific/agroecological) is dependent on practical community involvement. Practical community involvement can be achieved via dialog and community-based field trials. First, community dialog should start early, with collective problem identification, assessment of local solutions and avenues for their improvement, culminating in the co-creation of acceptable technologies/practices, which can best mitigate the identified challenge. Secondly, setting up appropriate field trials could reduce the consultative process. Farmers can observe firsthand how a technology is practiced and the associated outcomes. Such practical observations can hasten a farmer's decision making regarding adoption [197,198]. Such technology trials must have a clear community-accepted challenge or crisis, which the technology aims to address. It is reasoned here that such an approach could lead to reduced non-adoption and dis-adoption of sustainable agricultural practices.

*The need for national mapping of agroecological technologies*: Diverse outcomes are reported regarding the success of various agroecological practices. As pointed out by Ref. [13], and partly demonstrated by Ref. [199], in a no-till yield analysis, there is context specificity of the technologies. Such context specificity and lessons learned from adoption barriers are suggestive of a need for agroecological technology mapping. Here, it is suggested that such mapping should include at least the following seven key contexts: (1) locally acceptable agricultural challenges and the available local solutions (historical and current); (2) the geo-climatic conditions of a community; (3) crop suitability assessment; (4) locally adaptable crop cultivars; (5) the economic status of target communities; (6) landholding status (size and tenure); and (7) the available non-governmental players of the agricultural sector. Such mapping, where societal dispositions and the general biophysical characteristics of an area are determined, could help determine the feasible AE technology for piloting. Although not prescriptive, such mapping could enhance technology adoption. A slightly similar mapping was developed [200], but this focused only on climate-smart agriculture.

*Promotion of non-profit sustainable vegetable production*: Most sustainable agriculture practices have proven ecological benefits, including improving soil properties, promoting biodiversity and improving the water and nutrient status of soils. These known and non-monetary benefits should be a key message in sustainable agricultural practice dissemination. With the impacts of climate change affecting all aspects of agriculture, some farmers will be willing to adopt relevant sustainable agriculture practices just for the sake of participating in climate change mitigation. Monetary expectations from sustainable agriculture practice adoption usually take time [201] to be fulfilled. The concept of unfulfilled expectations [202] could, in part, stem from "misinformation" about the technology, for example, conservation agriculture in a soil and water improvement tool with known limitations in yield [203] compared to conventional agriculture, while promoting conservation agriculture as a solution to low agricultural productivity is not a truthful or prudent approach. "Unfulfilled promises" were found as a key reason for conservation agriculture dis-adoption in Malawi [204]. While research has shown that sustainable agriculture practices can simultaneously lead to poverty reduction and environmental protection [205], poverty reduction effects may take longer to materialize and should rarely be used as a positive reason for embracing a technology.

## 8. Conclusions

Vegetable production impacts natural resources in various ways, and its practice needs to be integrative and embrace the principles of sustainability. Irrespective of the criticisms and doubts regarding the workability of sustainable agriculture practice, there is ample evidence of positive outcomes from such practices for vegetables. By reviewing sustainable vegetable production from other regions (and comparing these cases with southern Africa), it is evident that sustainable vegetable cultivation has a key role in both the current and potential future agroecosystems. Like most approaches in integrated natural resource management, sustainable vegetable production can be adequately achieved with community involvement. The suggested framework for the promotion of sustainable vegetable production practices is community centered and can be adapted to other sustainability-related interventions in broader agriculture, fisheries and forest protection. Future predictions of changes to the agroecological system are bleak, especially for southern Africa, where land scarcity and erosion of the genetic diversity of vegetables are expected. With a constant need for land space on the planet due to unabated human population growth, it is imperative to promote various agroecological approaches to all stakeholders, of which rural communities (who are the most impacted during drastic agroecological changes) are key. The erosion of genetic resources can be mitigated using rich traditional knowledge from such communities. These communities, as custodians of indigenous vegetable species, will need to interact with experts in the prioritization and promotion of future adaptation strategies, including vegetable breeding. The sustainability of vegetable production,

and of agricultural resources in general, will require a holistic approach to the protection of agroecosystems.

**Author Contributions:** D.M.M. and H.G.: methodology and validation; D.M.M. and H.O.: writing—original draft preparation; D.M.M., H.G. and H.O.: writing—review and editing; L.F., S.M., H.K. and T.S.: supervision; H.O., H.K. and T.S.: project administration. All authors have read and agreed to the published version of the manuscript.

**Funding:** This study was funded by the "Establishment of a Sustainable Community Development Model based on Integrated Natural Resource Management System in Lake Malawi National Park (IntNRMS) Project" under the Science and Technology Research Partnership for Sustainable Development (SATREPS) program provided by the Japan Science and Technology Agency (JST) and Japan International Cooperation Agency (JICA) from 2020 to 2024 (JPMJSA1903). This work was also supported by JSPS KAKENHI Grant Number JP20K06351 and research programs of the Tokyo NODAI Research Institute, Tokyo University of Agriculture.

**Institutional Review Board Statement:** Not applicable.

**Informed Consent Statement:** Not applicable.

**Data Availability Statement:** The data used in this paper were acquired from the literature. For any queries regarding the source of any of the information presented here, interested experts can contact the corresponding author directly through e-mail.

**Acknowledgments:** This paper benefited greatly from observations made at a community vegetable garden in Chembe village, Cape Maclear, Mangochi, Malawi. The role played by this small garden in supporting the community's vegetable needs inspired a desire to study vegetable production further. John Banana Matewere (the leader of this community initiative) is acknowledged for providing some background information regarding this community garden. This paper also benefited greatly from discussions with IntNRMS project members, especially those of the Agriculture Resource Management Group of the project. Atsuko Fukushima (Ehime University) provided administrative support throughout the research process. This research would not have been completed without their help.

**Conflicts of Interest:** The authors declare no conflict of interest.

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
