# Peer review of "The Sustainable Niche for Vegetable Production within the Contentious Sustainable Agriculture Discourse: Barriers, Opportunities and Future Approaches"

_sustainability, doi:10.3390/su15064747_

Round 1

Reviewer 1 Report

The paper has a very well structured theoretical background and detailed conceptual overview. It does a very good job integrating the various scientific approaches facilitating integrated resource management under broader agroecology, which the current Climate-smart Agricultural (CSA) Innovation practices. The literature review is very well cited. The methodological approach that compares in very detailed, sustainable vegetable agricultural production practices under three major agroecology (i.e. Europe, China, and South Africa) is very noble. The comparisons cover all the essential themes or factors under sustainable vegetable production- agroforestry, conservation agriculture, crop rotation, green manure, mulching, irrigation, and others. Finally, it also addresses the critical barriers and opportunities for sustainable agriculture practices. The critical barriers to adoption cited include knowledge related, labor aspects, income related issue, farmland, equipment issues, crop-based factors, extension and policy, which are all crucial especially in regions such as Sub-Sharan Africa. Finally, it stresses the importance of preserving indigenous, which are basis of genetic biodiversity.

Author Response

Dear Reviewer,

Thank you very much for your review.

Find attached our response

Authors

Reviewer 2 Report

Dear Authors, 
Congratulation for your manuscript.

Please, see attached my comments and suggestions.

I hope they be useful.

Best

Author Response

Dear Reviewer,

Thank you very much for your review.

Attached here is our response regarding the edits made.

Authors.

Reviewer 3 Report

General remarks

The article offers a review from the literature on the application of agroecology in different regions highlighting barriers and opportunities and lessons that can be learned. The article is interesting and well argued, with an exhaustive collection from the literature. However, the organisation of the exposition is not always clear and sometimes it becomes difficult for the reader. Furthermore, in several parts the text is inaccurate.

The article sometimes presents acronyms that are not before presented and explained, such as in Table 1 (CA, CR, PM, GM..) or row 617 “AE”, or row 88 “SAP”. An acronym should always be explained the first time it appears in the text.

In several parts, the text is inaccurate and careful re-reading is recommended. In Table 1,2, e 3 the punctuation and use of upper- and lower-case letters seems confused and not checked. Other examples are rows 143, 188, 293 etc.

The terminology is also not always adequate: sometimes it is too complex, sometimes it is too journalistic (e.g. 'Fear of the decreased income'). Terms are not always stated in the same way, e.g. Sub-Saharan Africa or Southern Africa and it would be more useful to use an unambiguous indication.

Specific remarks per chapter

The introduction chapter is too long and does not highlight the architecture of the article, which is very complex. It is necessary to introduce the structure of the article to enable the reader to follow the exposition more easily. Perhaps it should be made clearer that the survey aims to support these practices for vegetables in Africa, in the light of the experiences of two comparative areas, Europe and China.

I recommend adding two points at the end of the introduction: the first clearly stating the purpose of the article (what it intends to  present); the second is an outline of the article presenting the different parts (e.g. the first chapter presents materials and methods, the second focuses on the reasons why it is necessary to practice sustainable agriculture for vegetables, the third presents the barriers and opportunities of adopting agroecology; in particular this chapter provides an articulation for the three selected areas, and the lessons learnt that can help the adoption of agroecological practices in Africa. The final part offers a proposal on the main keys to encourage the adoption of these practices).

A chapter indicating materials and methods is missing. This is a major criticism for the reader. From row 61 to 125 the text could be used for a specific chapter presenting the methodological choices: which practices were chosen and why, which areas and which bibliographical references. The absence of this chapter is very negative and the weakest point of the article.

The new chapter should well highlight the objective of supporting agroecological practices for vegetables in Africa, starting from the experiences of the two comparison areas, Europe, and China. The chapter should also explain why the authors consider such different areas to be comparable and thus how European lessons can be transferred to Africa. Furthermore, it should also explain why the topic of public support, which is actually very relevant, has not been considered.

Chapter 2 does not need all the sub-parts (2.1, 2.2, 2.3...) and can be better read as a homogeneous text.

Chapter 3 could provide a summary framework of the different techniques considered and the three selected areas in a synoptic form.

The chapter does not consider the public support; but at least in Europe the CAP, via both first pillar and rural development, is a very important factor. Here there is only a very quick mention in the part on China. Perhaps this is not enough.

Why is precision agriculture only examined in China? This detracts from the systematic nature of the exposition and leads to more doubts than anything else.

It is advisable to always use the same wording and order for the selected practices: e.g. intercropping and irrigation for China are reversed compared to the other parts; in the case of Africa the terms on row 374 is not consistent with that used for Europe and China.

Paragraph 3.4 Malawi study as a case point: it is not clear what it adds to the discussion. It is suggested to be removed.

Chapter 4 presents figure 3 as a summary on the barriers encountered, but without explaining it or introducing it. This is confusing. I recommend using the figure 3 after tables 2,3,4, to use it as final summary of the part. I also recommend making a graph also by region, and not only by item/barrier.

The part of the chapter presenting Tables 2,3 and 4 should be placed in a dedicated sub-section (e.g. 4.1 - A synoptic representation of the barriers surveyed by area) and it should have a text that adequately introduces and presents the three tables, even briefly.

The text from row 455 to 471 is not clear: it seems out of place and does not provide any key to reading Tables 2,3,4,. Can it be deleted?

The current paragraph 4.1 (which should become 4.2) offers a lot of detail. The question is whether it is necessary and whether the placement is correct. In fact, indicating the botanical name makes it very difficult to read. If, as it seems, it is an in-depth analysis on a specific topic, could this part be placed (if necessary) at the end of the chapter and after the analysis of the lessons learnt?

Paragraph 4.2 would perhaps be better placed immediately after the analysis of barriers and Tables 2,3,4. Moreover, if I understand correctly, it presents only the lessons learnt from Europe and China that are addressed to Africa.  In this case it should be clearly stated in the title of the paragraph and introduced with a short text. Or are lessons learnt from African experiences missing?

Also, in the case of Chapter 5, there is too much sub-paragraph articulation. Is this necessary? The different parts of the text (role of local communities; specificity of contexts, non-monetary benefits) are clear even without the paragraphs.

Author Response

Dear reviewer,

Thank you very much for your useful comments. Your comments have really improved our manuscript. Attached here are specific responses to your comments.

Authors

Round 2

Reviewer 2 Report

Dear Authors, 

Thank you for the new version of the manuscript.

Best regards

Author Response

Dear Reviewer,

Thank you for your comments on our manuscript.

Kind regards

Hiromu Okazawa (on behalf of authors)

Reviewer 3 Report

I really appreciate the work done. I consider this new version suitable for publication, after a final check on the formatting of the article.

For example, I suggest a check of all chapter and sub-chapter titles (row 39 244, 300, 362 vs row 53, 432, 447, 480, 512, 555) and their order (147 vs row 224). It would be important to have a similar presentation of the different parts, in fact.

Figure 3 also shows some inhomogeneities, both in the colour of the different balloons (or do the different colours mean something? if so, this should be explained) and in the horizontal balloon at the bottom.

 These are very minor observations. The article is very positive

Author Response

Dear Reviewer,

Thanks for your comments on our manuscript.

Attached is our response

Hiromu Okazawa (on behalf of authors)
